# Monitoring Volatile Organic Compounds and Aroma Profile of *Robinia pseudoacacia* L. Honey at Different Storage Temperatures during Shelf Life

**DOI:** 10.3390/foods12163105

**Published:** 2023-08-18

**Authors:** Sara Panseri, Federica Borgonovo, Marcella Guarino, Luca Chiesa, Maria Lucia Piana, Rita Rizzi, Michele Mortarino

**Affiliations:** 1Department of Veterinary Medicine and Animal Sciences, Università degli Studi di Milano, Via dell’Università 6, 26900 Lodi, Italy; sara.panseri@unimi.it (S.P.); luca.chiesa@unimi.it (L.C.); rita.rizzi@unimi.it (R.R.); 2Department of Environmental Science and Policy, Università degli Studi di Milano, Via Celoria 2, 20133 Milano, Italy; federica.borgonovo@unimi.it (F.B.); marcella.guarino@unimi.it (M.G.); 3Piana Ricerca e Consulenza S.r.l., Via Umbria 41, Castel San Pietro Terme, 40024 Bologna, Italy; pianaricerca@pianaricerca.it

**Keywords:** *Robinia pseudoacacia* honey, volatile organic compounds, SPME-GC-MS analysis, aroma profile, electronic nose, storage temperature

## Abstract

Bee honey has different volatile organic compound profiles that depend on the botanical origin and the state of conservation and which are mainly responsible for its specific aroma. During honey storage, the profile of these molecules and other indicators, such as 5-hydroxymethylfurfural and the diastatic index, can change depending on temperature and time. This study analyzed the variations that these parameters in acacia honey stored at three different temperatures for a total period of 550 days, using gas chromatography coupled with mass spectrometry and an electronic nose equipped with 10 different sensors. The results confirm that the composition of acacia honey varies over time due to both the reduction in the concentration of volatile molecules (e.g., formic acid, a natural acaricide) and the increase in compounds resulting from heat-dependent degradations (e.g., 5-hydroxymethylfurfural). This study supports the usefulness of the electronic nose for the early detection of aromatic alterations in honey subjected to high-temperature storage.

## 1. Introduction

The composition of honey is greatly variable and complex as it depends on several factors, such as botanical source, nature of the soil, climate, colony physiology, techniques employed by beekeepers, honey harvesting, and post-collection processing. Honey contains about 200 substances, mainly sugars, water, proteins (enzymes), organic acids, vitamins, minerals, pigments, phenolic compounds, a large variety of volatile compounds, and solid particles derived from honey harvesting [1].

Honey is considered a stable product, normally not attacked by microorganisms that cause alterations in most food products. However, it undergoes an aging process that involves physical, chemical, and organoleptic alterations, namely, loss of volatile substances, darkening of the color, modifications of the sugar composition, decrease in the enzymatic charge, and increase in 5-hydroxymethylfurfural (HMF) [2,3]. 

The volatile organic compounds (VOCs) constitute minor components of honey and belong to different chemical families, such as hydrocarbons, aldehydes, alcohols, ketones, acids, esters, benzene and its derivatives, norisoprenoids, terpenes, and cyclic compounds [1,3]. Although they represent a very small fraction of the overall honey composition, they are different for each species of honey and contribute to characterizing the aroma of the product [4]. As for aroma components, most kinds of honey naturally contain several compounds that are also constituents of organic acids and essential oils used in apiculture for their acaricidal activity, such as formic acid and thymol, respectively [5,6,7]. The volatile compound’s profile can change during storage as a result of the temperature and the period to which it has been exposed [8].

When honey is heated or stored for a long time, pentoses and hexoses decompose and form undesirable compounds such as furans, mainly furfural and 5-hydroxymethylfurfural (HMF), that represent the main degradation products of sugars, and their occurrence in foods is usually related to non-enzymatic browning reactions, i.e., Maillard reaction, sugar degradation in an acidic medium, and caramelization. Although not dangerous for humans, HMF is used as an indicator of honey conservation quality, and its concentration must be below 40 mg kg^−1^ [9]. Indeed, HMF increases gradually in all kinds of honey during storage, but its production is accelerated with moderate heating (35–40 °C) for some months or heating at high temperatures for a short time [10]. Beyond the HMF value, also the diastatic index is another important indicator of the freshness and chemical degradation that honey undergoes during storage; in the presence of water, the diastasis enzyme splits the oligosaccharides into simpler compounds, and its activity may be reduced during storage, and modified by denaturation, brought about by heating [11]. 

The diastatic activity of honey is influenced by the storage and heating conditions and, therefore, represents an indicator of the state of freshness and overheating of honey, considering a minimum of eight Diastase Numbers (DN) expressed in Schade units to be valid, although diastasis has a wide natural variation [9,11].

The analysis of the volatile compounds detected with different methodologies allows for assessing the aroma and the floral origin of the honey, and also, the appropriate conservation of the product and different chemical analyses can be combined with the use of an electronic nose (e-nose) to analyze the aromatic compounds present in honey [12,13]. The e-nose can mimic the human sense of smell, and the device consists of multiple sensors that can modify their electrical characteristics in the presence of various volatile compounds. These are non-specific sensors and, therefore, unable to recognize the various substances and return values that are an olfactory imprint. The e-nose technology, which works as a human olfactory system, is widely used for different applications in many industrial production processes that range from environmental monitoring (air contaminants detection) to food safety and medical applications [13]. In regards to the food industry, the e-nose is one of the best solutions to monitor and analyze the quality of food and agro-products and to assess the freshness through shelf-life investigation, as reported in this review [14]. During the storage period, the odor emitted from food products is an important indicator of quality and product conformity, also considering the potential of an electronic nose to perform odor assessment with a minimal cost, thus providing a rapid and predictive response compared to the laboratory methods which require more time [15]. Regarding honey, in the past years, e-nose technology has been mainly used for the classification and assessment of botanical or geographic origin [16,17,18].

In the present study, it is assumed that the application of a combined analytic approach focused on VOCs and odor analysis could help the comprehensive evaluation of honey quality and freshness in time. Thus, the aims of this study were first set to investigate the trend of VOCs of *Robinia pseudoacacia* L. honey samples during storage at different temperatures by HS-SPME-GC-MS. A second objective was to assess the feasibility use of the e-nose to recognize and analyze the quality of the same honey in order to confirm this technique to be involved in rapid quality control.

## 2. Materials and Methods

### 2.1. Honey Sampling and Timing of the Analysis

*Robinia pseudoacacia* honey was obtained from the apiary of the Veterinary Faculty situated in Lodi, in the Po Valley (87 m a.s.l.). After complete capping, honey supers were taken from the hives, and honey was readily extracted to a 50 kg settling tank. After two weeks, 15 glass jars of 1 kg each were filled with the settled honey. Then, five jars were stored in each of three thermostats with temperatures set at 15 °C, 25 °C, and 35 °C, respectively. In this regard, the temperatures of 15 °C (ST15) and 25 °C (ST25) have been considered as representative of the most common storage conditions, whereas the temperature of 35 °C (ST35) was selected as extreme storage conditions at which the biochemical processes are accelerated.

Just before the storage on Day 0 (T0), five aliquots of 35 g of honey were collected from the tank, and this time point was set as T0. Then, during storage at Day 70 (T1), Day 130 (T2), Day 190 (T3), Day 250 (T4), Day 310 (T5), Day 380 (T6), Day 440 (T7), and Day 550 (T8), 35 g of honey was taken from each jar. Then, for each sample, the VOCs profile was determined by SPME- GC/MS analysis, and the aromatic profile was evaluated by an e-nose equipped with a sensor array made up of 10 chemical sensors. In order to control the honey quality, the diastase activity at each storage temperature was determined on pooled samples obtained from 10 g aliquots pipetted from each of the five jars at T1, T2, T4, and T8. 

### 2.2. HS-SPME of VOCs

All the samples were prepared by weighing 5 g of honey in a 20 mL glass vial fitted with a cap and equipped with silicon/PTFE septa (Supelco, Bellefonte, PA, USA) and by adding 1 mL of the internal standard solution (IS) in water (1,4-cineol, 1 μg/mL, CAS 470-67-7) to check the quality of the fibers. At the end of the sample equilibration period (1 h), a conditioned (1.5 h at 280 °C) 50/30 μm Divinylbenzene/Carboxen/polydimethylsiloxane (CAR/PDMS/DVB) StableFlex fiber (Supelco, Bellefonte, PA, USA) was exposed to the headspace of the sample for the extraction (180 min) by CombiPAL system injector autosampler (CTC Analytics, Zwingen, Switzerland). The fiber and the time of extraction used in this study were selected after a preliminary study [19,20,21]. The equilibrium and extraction times were 15 min and 180 min, respectively, by using CAR/PDMS/DVB fiber. The temperature of 25 °C was selected in order to prevent possible matrix alterations (oxidation of some compounds, particularly aldehydes and furans).

To keep a constant temperature during analysis, the vials were maintained on a heater plate (CTC Analytics, Zwingen, Switzerland). As demonstrated in other research in which the VOCs profile of food is investigated, the use of high extraction temperature can lead to ex novo formation of volatile compounds or to the production of artifacts [22,23].

### 2.3. Gas Chromatography–Mass Spectrometry Analysis of VOCs

HS-SPME analysis was performed using a Trace GC Ultra (Thermo-Fisher Scientific, Waltham, MA, USA) Gas Chromatograph coupled to a quadrupole Mass Spectrometer Trace DSQ (Thermo-Fisher Scientific, Waltham, MA, USA) and equipped with an Rtx-Wax column (30 m; 0.25 mm i.d.; 0.25 μm film thickness, Restek Corporation, Bellefonte, PA, USA). The oven temperature program was as follows: from 35 °C, hold for 8 min to 60 °C at 4 °C/min; then, from 60 °C to 160 °C at 6 °C/min; and finally, from 160 °C to 200 °C at 20 °C/min. Carryover and peaks originating from the fiber were regularly assessed by running blank samples. After each analysis, fibers were immediately thermally desorbed in the GC injector for 5 min at 250 °C to prevent contamination. The injections were performed in splitless mode (5 min). The carrier gas was helium at a constant flow of 1 mL min^−1^. The transfer line to the mass spectrometer was maintained at 230 °C, and the ion source temperature was set at 250 °C. The mass spectra were obtained by using a mass selective detector with the electronic impact at 70 eV, a multiplier voltage of 1456 V, and by collecting the data at a rate of 1 scan s^−1^ over the *m*/*z* range of 30–350. Compounds were identified by comparing the retention times of the chromatographic peaks with those of authentic compounds analyzed under the same conditions when available. The identification of MS fragmentation patterns was performed either by comparison with those of pure compounds or using the National Institute of Standards and Technology (NIST) MS spectral database. Volatile compound measurements from each headspace of honey extracts were carried out by peak area normalization (expressed in percentage).

### 2.4. Diastase Activity DeterminationVOCs

The diastase activity of the pooled honey samples at each storage temperature at T1, T2, T4, and T8 time points was evaluated with Phadebas [24,25], and the results were expressed as Diastase Number (DN) in Schade units.

### 2.5. E-Nose Measurements

Analyses were performed using a portable Electronic Nose (PEN3) from Winmuster Airesense Analytics (Schwerin, Germany) which is composed of 10 metal oxide sensors (MOS). Table 1 lists all the sensors used and their applications. 

The sensor response is expressed as resistivity (Ohm). As evidenced in the scientific literature, MOS sensors are the most suitable ones compared to polymer sensors for food analysis. In fact, these sensors work at high temperatures and are not sensitive to humidity [18].

All the samples were prepared by weighing 3 g of honey in small sealed glass vials with a capacity of 40 mL in order to create the correct head space. During the analysis, the vials were kept in a thermostatic water bath at a constant temperature of 40 ± 2 °C to prevent the effects of thermal fluctuation.

The analysis protocol was defined by setting up the e-nose parameters (flow rate, duration of measurement, etc.) according to the manufacturer’s instructions. The analysis of each sample lasted 100 s, which was enough for the sensor’s signals to reach a stable value in seconds.

The set of signals derived from the electronic nose during the analysis took the form of a pattern that was analyzed using WinMuster^®^ (version 1.6.2.17 May 2014, Airsense Analytics GmbH).

### 2.6. Statistical Analyses

The starting datasets included 120 records relative to chemical measurements and 600 records relative to E-nose responses measured during the last 5 s when sensors’ signals were stable. All data editing and analyses were conducted using Statistical Analysis System (SAS 9.4) software. A variance analysis was carried out using PROC MIXED in order to investigate the effect of temperature and time on the VOC classes and HMF (see Results). The mixed model included the fixed effects of time (8 levels), storage temperature (3 levels), the interaction between time and storage temperature (24 levels), and the random effect of the jar (5 levels). For VOC compounds and HMF, results are presented as least squares means, which were compared using the PDIFF option. Significance was indicated at *p* ≤ 0.05.

Canonical discriminant analysis was applied in order to find linear combinations of quantitative variables that provide maximal separation among groups. In our study, the classification variable, i.e., the 3 storage temperatures and the E-nose responses (W1A, W5B, W3A, W6B, W5A, W1B, W1C, W2B, W2C, and W3B) were analyzed to derive canonical variates. These are linear combinations of the E-nose responses that contained the highest possible multiple correlations with each storage temperature within each time. The variable defined by the linear combination was the first canonical variable. The second canonical correlation was obtained by finding the linear combination uncorrelated with the first canonical variable that had the highest possible multiple correlations with the classes. Canonical loadings measure the linear correlation between an original variable (i.e., ae-nose responses) and the canonical variate; they can be interpreted as the relative contribution to each canonical variate. PROC CANDISC was used to derive canonical variates, canonical loadings and canonical correlations. A *p*-value of 0.05 was used to verify the significance of all statistical tests.

## 3. Results

### 3.1. Analysis of VOCs, HMF, and Diastasic Activity

The volatile profile of acacia honey samples used in this investigation, expressed as the average percentage throughout the storage period for each temperature, is shown in Table 2.

Overall, one hundred and one compounds were identified, but among these, only seventy were present in honey at T0. The identified VOCs belonged to nine different major chemical classes: aldehydes; alcohols; sulfur compounds; nitrogen compounds; free fatty acids; furans; hydrocarbons; ketones; and terpenes. In addition, since HMF is an indicator of honey quality during storage, it was considered separately and expressed as mg kg^−1^ of honey.

Before storage, 83.17% of the compounds belonged to the classes of free fatty acids (52.32%) and alcohols (30.85%), in which the most abundant compounds were acetic acid (29.79%) and ethanol (20.55%). Furthermore, formic acid was detected as a relatively (2.22% at T0) abundant component of the class of free fatty acids, with a temperature-dependent decrease during storage.

Other classes of chemical compounds have been identified in smaller quantities (approximately 3% each): hydrocarbons (3.51%); ketones (3.07%); aldehydes (2.98%); and terpenes (2.96). Seven out of 18 alcohols have not been detected in the fresh product and developed subsequently. Only three furanic compounds were found in fresh honey, among which the furfural showed the highest amount. Among the aldehydes, nonanal and benzaldehyde are the most abundant compounds, while the aldehyde lilac lacks in the fresh honey and is formed during the conservation period. In regards to terpenes, the amount of hotrienol, linalool, and trams-linalool is noteworthy. Among the miscellaneous compounds, the monoterpenoid thymol showed a rather stable proportion during storage.

Table 3 shows the LSMeans of HMF and of the nine chemical classes for each storage temperature, i.e., the average amount during storage at each temperature adjusted for the factors included in the model. Significant differences (*p* < 0.05) were found between the averages estimated at the different storage temperatures for five classes of chemical compounds (ketones, free fatty acids, furans, nitrogen, and sulfur compounds). 

A decreasing trend from ST15 to ST35 was found for free fatty acids (30.44, 26.63, and 24.22%) and nitrogen compounds (2.34, 1.64, and 1.24%). Conversely, an increasing trend is notable for the classes of furans and sulfur compounds, whose average amounts are significantly higher at ST35 than at ST25 and at ST15. Similarly, the highest number of ketones is detected at ST35.

The content of aldehydes, terpenes, and alcohols at ST35 is significantly different from that found at both ST25 and ST15. In particular, at ST35, the aldehyde and the terpene contents are higher, while the alcohol content is lower than at ST25 and at ST15. Finally, the content of HMF significantly differs at the three temperatures, with an increasing trend of LSMeans from ST15 to ST35.

When considering the amount of each chemical class at different storage temperatures, there is an increasing trend from ST25 to ST35 for furans, ketones, and hydrocarbons, a decrease in alcohols and sulfur compounds and a fluctuating trend for the remaining classes (Figure 1A). The furans level is significantly higher at 35°C throughout the storage period compared to the other storage temperatures, and significant differences were also found for the samples stored at ST15 and ST25 after 190 days. It is possible to see a similar trend for ketones, still with significantly higher levels at ST35, while the level of ketones at ST15 undergoes a significant increase compared to the other storage temperatures after 380 days. Regarding the increase in hydrocarbons over time, there is no difference among the storage temperatures, except for the period between 310 and 440 days when the level of ketones at ST35 is significantly lower.

The alcohol content at ST15 and ST25 shows a moderate increase up to 180 days, a plateau and a decrease after 310 days, but no significant differences at the individual controls; on the contrary, the performance of this class at ST35 C is significantly different, and there is a constant decrease throughout the storage period.

The trend of acids descends to ST15, ascends to ST35, and remains almost constant at ST25, with significant differences between the storage temperatures up to the fifth control (310 days). 

The quantity of sulfur compounds clearly falls at ST35 during storage and presents peaks and depressions at ST15 and ST25 with significant differences in the controls between the three temperatures up to the sixth control, after which there is a decrease with values almost similar.

The class of nitrogen compounds shows similar trends for each storage temperature with a peak at 180 days and a sudden decrease up to 310 days; subsequently, for the samples stored at ST15, a very high and significantly different content was detected at the end of storage in comparison to those stored at higher temperatures.

During the storage period, an increase in terpenes is noted at the second and sixth controls for all three temperatures. For the class of aldehydes, there is a decrease up to the fourth control, in particular, for the samples stored at ST35, followed by a slight increase for all storage temperatures.

The analysis of HMF content and diastatic activity over time (Figure 1, Panel B, and Table 4, respectively) indicates that honey does not meet the requirements for food consumption after 130 days (T2) if stored at ST35 and after 18 months (T8) if stored at ST25. On the other hand, these two parameters remain stable and at optimal levels after the 18-month storage at ST15. 

### 3.2. Canonical Discriminant Analysis

Table 5 shows the canonical loadings of the 10 sensors on the first two canonical variates (Can1 and Can2), the relative percentage of variance, and the canonical correlations between the original variables and the canonical variates for each time. 

The first canonical variate relative to T1 and T7, respectively, accounted for 85% and 72% of the variance, whereas for the other times, the magnitude of this value is greater than 95%. At each time, the canonical correlations were very high and significant (*p* < 0.0001), being greater than 0.95 for the first canonical variate and with a range from 0.72 to 0.97 for the second. At T1 and T8, the first canonical functions were dominated by large positive loadings from sensors for aromatic compounds (2-W5S, 3-W3C and 1-W1C) and by negative loadings from 9-W2W sensors reactive to nitrogen oxides and ozone. On the contrary, this latter sensor showed positive loadings at T6 and T7. Large positive loadings were found at T2, T5, and T6 for sensor reactive to sulfur compounds and terpenes (10-w3S) and at T3 and T4 for sensor selective to methane (6_W1S).

Figure 2 shows the scatter plots of the canonical scores of the samples for the two canonical variates at each time. 

At T1, the honey samples grouped on the basis of the storage temperature but were not clearly separated as in subsequent times. From T2 to T7, the group stored at ST35 was separated from the other two based on the first canonical variable, and at the end of the storage period, it was hardly distinguishable from the group at ST25. The second canonical variable is instead the discriminating factor for the groups at ST15 and ST25 that appear clearly separated starting from T4.

## 4. Discussion

The analysis of VOCs is a useful tool to characterize unifloral honey since VOCs are the main factors responsible for aroma. Some components derive from the nectar of the flowers and contribute to the identification of the botanical origin and to the characterization of the aromatic profile of the honey. These components are, therefore, considered floral markers of particular honey. However, it is difficult to identify unique markers for honey with the same botanical origin since the composition of pollen or nectar is influenced by other factors such as bee species, geographic area, season, mode of storage, and harvest technology and methods of extracting the volatile compounds [3,26]. Therefore, the aromatic profile of honey from the same botanical origin is characterized by different chemical compounds.

In HS-SPME analysis of VOCs, the fiber type, as well as extraction temperature, represent important variables since they can influence the obtained VOC profile as well. In the present study, CAR/PDMS/DVB fiber was selected as it represented a useful tool to obtain a comprehensive wide-range VOC profile for foods such as honey and other bee products. The number of VOCs identified in fresh honey increased from seventy to about one hundred compounds during storage and regardless of storage temperature. Detection of compounds in fresh honey that are not present during storage can be explained by the presence of yeasts and microorganisms and by the different metabolic pathways leading to new products and new volatile compounds [1,8].

Acetic acid and ethanol are among the main volatile compounds present in honey of different geographical and botanical origins. The first compound could be produced by bee metabolism, while the high percentage of the second one can be attributed to the presence of yeasts that transform carbohydrates into alcohols [27. Moreover, the large amounts of these two compounds depend on the extraction method since the CAR/PDMS fiber shows a great affinity for compounds with low molecular weight [27,28]. 

The acacia honey may be characterized on the basis of its content of lilac aldehyde A [20,29], hexanal [3], and nonanal and its homologs, such as heptanal, octanal, and decanal [30]. Our analyses only confirmed the presence of nonanal and decanal. Some terpenes are responsible for the specific aroma, such as hotrienol, linalool, and its oxide, cis-linalool oxide, was previously detected in acacia honey [29]. All these terpenes were identified in this research except cis-linalool oxide since we found its enantiomer, trans-linalool oxide, which is considered a floral marker of chestnut honey.

Several volatile compounds in our sample were also reported in the acacia honey of Romania [31,32] and in two Italian honeys collected in mountainous areas of Lombardy [19] and in Southern Italy [33]. On the other hand, some compounds, e.g., trans linalool oxide, were present only in Italian acacia honey and not in the Romanian one. This finding could support the analysis of specific VOCs as potential markers of geographical origin.

The diastatic index and the HMF value are important indicators of the freshness and chemical degradation that honey undergoes during storage. In the presence of water, the diastasis enzyme splits the oligosaccharides into simpler compounds, and its activity may be reduced during storage and modified by denaturation, brought about by heating [11]. The degradation of the diastase observed in our sample is in agreement with the estimated half-life of the enzyme during honey storage, which is approximately 17 months at 23–28 °C and 100 days at 35 °C [10]. When the diastatic activity was measured in two types of honey stored at 35 °C, the values below the limit threshold of 6 months for floral honey were found, while the values below 8 DN were detected at 12 months in only 3 samples out of 15 of honeydew honey, which in itself is characterized by a high content of diastase [34]. The trend of HMF content at three different temperatures confirms that honey stored at 15 °C maintains the characteristics of freshness for durability of 18 months. At temperatures equal to or higher than 25 °C, honey degrades, particularly quickly, if stored at 35 °C. The furans content over time at the three different storage temperatures follows the same trend of HMF, with a dramatic increase at the highest temperature. Heat can cause chemical changes in honey, among which the ones due to the reaction of Maillard are noteworthy. Indeed, during the reaction at pH 7 or below, the Amadori product undergoes enolization with the formation of HMF or furfural when hexoses or pentoses are involved, respectively [2]. The average furfural content at 35 °C is most likely attributable to the degradation of sugars, while the small percentage in fresh honey may be transported from comb wax [35].

Ketones increased during storage at all three temperatures, in contrast to what was reported by other authors [36]. Among ketones, the most represented compounds were acetone at 15 °C and hydroxyacetone at 35 °C: the former is normally present in acacia honey [12], while the latter derives from the Amadori product during the Maillard reaction [2].

The reduction in alcohols, observed at 35 °C during storage, was observed in Brazilian honey over six months and was probably due to volatilization, ester formation, and oxidative processes [36]. On the contrary, at 15 °C and 25 °C, the trend shows a parabolic shape with a limited loss of these compounds at the end of the conservation period. This is in agreement with results from previous research reporting that 87% of 33 honey samples stored at 20 °C for one year had an increase and a subsequent decrease in ethanol, which is the largest percentage compound [37]. 

During storage, the trend of hydrocarbons does not show significant differences at three temperatures; however, hexane and phenylbutene have been found in both samples collected in Poland and South Africa [3,4], the latter only at 35 °C. These compounds can derive from changes caused by processes such as specific enzymes present in honey, high temperatures, and long storage periods [1] or can arise from beeswax, which has not been completely separated during harvest and processing [38].

The acid class included only non-aromatic acids, with the exception of benzoic acid. It seems that many non-aromatic acids are intermediate to the Krebs cycle or similar pathways and that they can be synthesized from nectar’s sugars both by the action of worker bee enzymes or directly [6]. Acids decrease during storage, especially acetic acid and formic acid: the latter compound is present in fresh honey due to its natural presence in plants that are visited by honeybees, and rapid decrease in its content in hive matrices is also reported after application as an acaricidal treatment against the mite Varroa destructor through removable strips [39]. It should be considered that acetic acid may also be produced in honey as a result of mild fermentation phenomena that can start at higher storage temperatures; this may have contributed to the concentration trend observed in the present study, which declined not as fast as it might have been expected during storage. Furthermore, the percentage of this chemical class at 35 °C appears more stable in time for some compounds compared to lower storage temperatures; for instance, at 35 °C, butenoic acid and 2-butenoic acids are present in higher percentages. Since these two acids are linked to a rancid aroma [3], they could be further indicators of the degradation of honey at high temperatures. The monoterpenoid thymol, another biologically interesting compound that is used in apiculture as a miticide, seems more persistent in honey than in organic acids. 

The changes in the VOCs during the storage described above are confirmed by the results of the discriminant analysis conducted on the e-nose sensors at various storage times. The loading coefficients suggest that the classes of compounds recognized by the e-nose sensors change at different times of the storage period without following a specific pattern. After all, honey is a very complex product that can modify its composition and properties during storage, and these variations can be attributed to two principal sources—compounds that are heat labile and may be destroyed and volatile compounds produced by non-enzymatic browning [3].

On the other hand, it can be inferred that the sensors can differentiate honey stored at different temperatures at any time. The separation of the groups at various times indicates a similarity in the response of the sensors of the two groups at 15 and 25 °C until the last detection when the honey at 15 °C clearly differs from the other two groups that seem to have similar characteristics. The discrimination of three groups throughout the storage period seems primarily related to the formation of HMF, which had values above the threshold value of 40 mg/kg^−1^ already at 70 days, if the honey was stored at 35 °C, and at 18 months if stored at 25 °C. 

The significant canonical correlation between the storage temperatures and the first and second canonical variates indicate that these latter explain the differentiation of the storage groups. Regarding the e-nose technique, it is less accurate than HS SPME and GC/MS but less expensive than the traditional analysis. It is useful, faster, and does not require qualified manpower. Results from the multivariate analysis are consistent with those from this study on the use of the e-nose in the food industry in order to discriminate and monitor the shelf life of fruit, vegetables, and animal products [40,41,42,43] and also to detect bacteria growth [44]. It can, therefore, be assumed that the e-nose can effectively recognize overheated honey from the first months after harvesting.

## 5. Conclusions

The quality assessment of bee acacia honey during shelf life can be effectively supported by a combined SPME-GC-MS and e-nose analytical approach; this last is a useful rapid tool for honey quality assessment. An electronic nose is a rapid technology to explore biological olfactory function; it was used to distinguish complex volatiles, which can reproduce the structure and principle of the olfactory sense. 

With the development of society, the importance of olfactory application in the food industry, environmental detection, and medical treatment is increasing. It is one of the indispensable tools for quality assurance and quality control, and its application to monitoring honey shelf life was demonstrated by the present study as a useful technique to control honey quality. E-nose is less accurate than SPME-GC-MS, but on the contrary, it is less expensive and faster than traditional analysis, and it does not require qualified manpower. Monitoring during storage at different temperatures shows that the VOCs profile at 15 °C is comparable with the one at 25 °C until 15 months of storage and begins to deviate from that at 25 °C only after 15 months. Further research could confirm that analysis by e-nose may be a cost-effective way to monitor the effect of overheating during storage in order to be applicable toward fraudulent practice detection as well.

## Figures and Tables

**Figure 1 foods-12-03105-f001:**
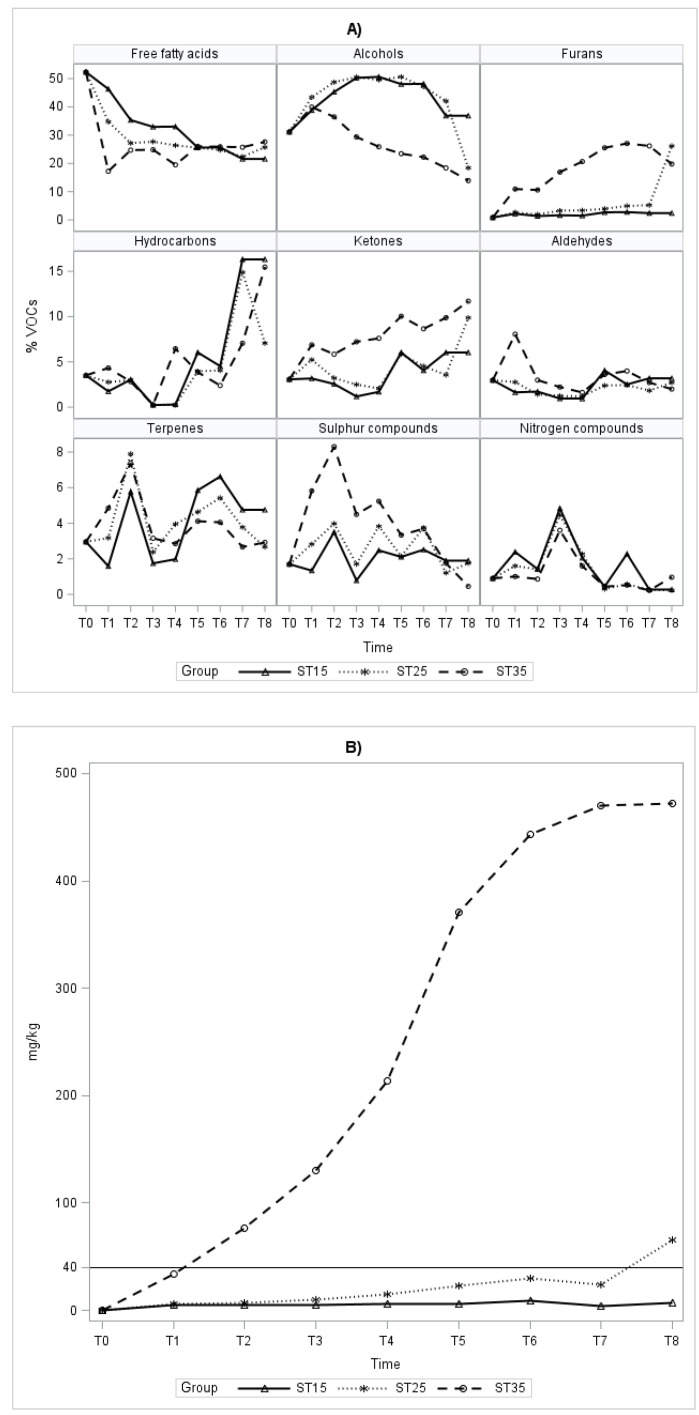
VOCs (panel (**A**)) and HMF (panel (**B**)) trend during the storage period for each storage temperature group. Least square means are reported from T1 to T8; raw mean of five aliquots is reported for T0.

**Figure 2 foods-12-03105-f002:**
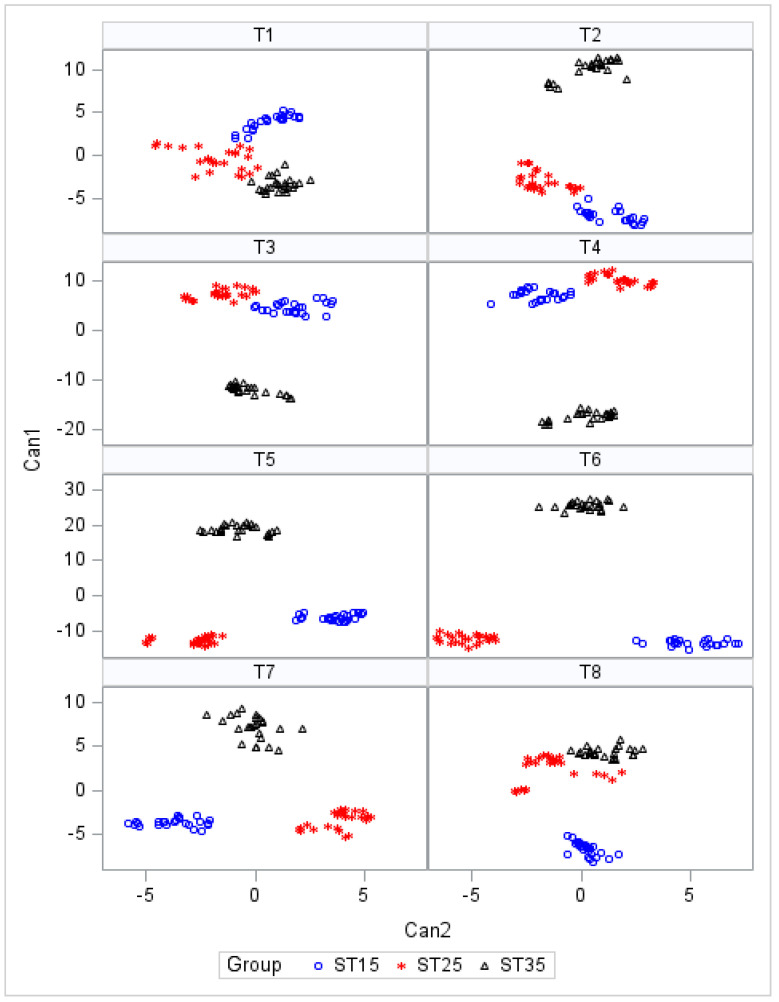
Score plots of honey samples at different storage temperatures and at each time.

**Table 1 foods-12-03105-t001:** Sensors and their applications in PEN3.

Number in Array	Sensor-Name	General Description	Reference
1	W1C	Aromatic compounds	Toluene, 10 ppm
2	W5S	Very sensitive, broad range sensitivity, react to nitrogen oxides and ozone, very sensitive to negative signal	NO_2_, 1 ppm
3	W3C	Ammonia, used as sensor for aromatic compounds	Benzene, 10 ppm
4	W6S	Mainly hydrogen, selectively (breath gases)	H_2_, 100 ppb
5	W5C	Alkanes, aromatic compounds, less polar compounds	Propane, 1 ppm
6	W1S	Sensitive to methane (environment) ca. 10 ppm, broad range similar to no.8	CH_4_, 100 ppm
7	W1W	Reacts to sulfur compounds (H2S 0.1 ppm). Otherwise sensitive to many terpenes and sulfur organic compounds, which are important for smell (limonene, pyrazine)	H_2_S, 1 ppm,
8	W2S	Detects alcohols, partially aromatic compounds, broad range	CO, 100 ppm
9	W2W	Aromatic compounds, sulfur organic compounds	H_2_S, 1 ppm
10	W3S	Reacts to high concentrations > 100 ppm, sometimes very selective (methane)	CH_4_, 10 ppm

**Table 2 foods-12-03105-t002:** Volatile compound profiles detected in honey samples stored at different temperatures.

COMPOUND	T0 a *	ST15 b **	ST25 b	ST35 b	COMPOUND	T0	ST15	ST25	ST35
**Aldehydes**					**Free fatty acids**				
Acetaldehyde	0.29	0.28	0.45	0.79	Acetic acid	29.79	20.01	18.11	16.2
2-Methylbutanal	nd	0.05	0.04	0.07	Formic acid	2.22	1.73	1.28	1.38
3-Methylbutanal	0.26	0.15	0.2	0.2	Propanoic acid	0.70	0.58	0.48	0.39
4-Methyl-2-pentanal	0.08	0.24	0.09	0.09	2-Methylpropanoic acid	1.72	1.7	1.41	0.99
2-Methyl-2-butenal	0.23	0.18	0.14	0.5	Pivalic acid	1.01	0.64	0.08	0.12
3-Methyl-2-butenal	0.39	0.62	0.38	0.65	Butanoic acid	0.86	0.66	0.63	1.42
3-hidroxy-2-butanal	0.43	0.53	0.35	0.43	3-Methylbutanoic acid	4.56	2.35	2.1	1.63
Nonanal	0.49	0.11	0.06	0.04	2-Butenoic acid	nd	0.03	0.05	1.89
Decanal	0.16	0.06	0.11	0.35	Pentanoic acid	0.85	0.22	0.08	0.13
Benzaldehyde	0.48	0.29	0.33	0.66	2,4 Hexadienoic acid	nd	nd	0.02	0.35
Lilac aldehyde A	nd	0.21	0.24	0.06	3-Methyl-2-butenoic acid	0.97	0.64	0.58	0.51
Lilac aldehyde B	nd	0.05	0.04	0.04	Hexanoic acid	6.55	1.28	1.05	1.08
Benzeneacetaldehyde	0.12	0.27	0.28	0.84	2-Ethyl hexanoic acid	1.74	0.39	0.45	0.49
Cinnamaldehyde	0.05	0.07	0.09	0.17	Octanoic acid	0.50	0.33	0.38	0.37
**Alcohols**					Nonanoic acid	0.86	0.23	0.25	0.35
Ethanol	20.55	28.04	28.65	13.79	Benzoic acid	nd	0.1	0.09	0.09
2-Methyl-3-buten-2-ol	0.78	1.03	1.37	1.35	**Terpenes**				
2-Methylpropanol	nd	0.68	0.73	0.26	Trans-Linalool oxide	0.25	0.49	0.74	1.66
2-Methyl-2-propanol	0.46	0.38	0.91	0.24	Linalool	0.72	0.77	0.75	0.32
1-Butanol	0.13	0.47	0.35	0.26	Pinocarvon	0.12	0.05	0.08	0.15
3-Methylbutanol	0.89	3.12	3.09	2.3	Hotrienol	1.23	0.51	0.81	0.7
1,7-Octadien-2-ol	0.85	1.29	0.99	0.62	4-Terpineol	nd	0.02	0.02	0.02
3-Methyl-3-butenol	2.46	4.2	4.34	3.95	Menthol	nd	2.52	2.07	0.94
4-Methyl-1-pentanol	nd	3.02	2.47	0.98	Terpineol	0.10	0.28	0.14	0.2
3-Methyl-2-buten-1-ol	2.68	4.24	4.53	3.06	Borneol	nd	0.02	0.04	0.03
3-Methyl-1-pentanol	nd	0.57	0.42	0.17	Epoxylinalool	0.16	0.25	0.27	0.17
3-Methyl-3-butenol	nd	0.04	0.04	0.02	β-damascenone	0.18	0.09	0.16	0.15
1-Hexanol	0.05	0.05	0.06	0.04	β-phellandrene	0.03	0.04	0.04	0.09
3-Hexenol	0.04	0.08	0.1	0.06	β-ionone	0.04	nd	0.01	0.01
2-Butoxy ethanol	nd	0.22	0.1	0.1	Terpen	0.13	0.08	0.09	0.29
Unknown	nd	0.06	0.1	0.62	Thymol	0.03	0.03	0.02	0.02
Benzenethanol	1.97	2.39	2.92	2.65	**Furans**				
Unknown	nd	0.03	0.05	0.06	Furan	0.13	0.4	0.17	0.14
Phenol	0.03	0.03	0.07	0.05	2-Methyl furan	nd	0.04	0.06	0.05
					2-Furan methanol	nd	0.05	0.05	0.06
**Hydrocarbons**					Furfural	0.66	1.79	3.91	16.1
Hexane	0.20	3.78	4.17	3.28	1-(2-Furanyl)-ethanone	nd	0.1	0.02	0.39
1,4-Pentadiene	nd	0.06	0.09	0.06	5-Methylfurfural	nd	0.17	0.85	2.41
2-Nitropropane Unknown	1.08 0.04	1.39 0.55	1.76 1.23	0.9 0.46	Dihydro-5-Methyl-2-3H-furanone	nd	0.12	0.77	2.1
Ethylbenzene	0.48	0.39	0.23	0.32	Dihydro-3-Methyl-2-3H-furanone	0.05	0.05	0.08	0.63
p-Xylene	0.37	nd	nd	0.03	2(5H)-Furanone	nd	0.22	0.32	1.8
o-Xylene	0.59	1.59	1.04	0.64	4,5-Dimethyl-2-furaldehyde	nd	0.01	0.04	0.8
m-Xylene	0.18	0.01	0.01	0.03					
Tetradecane	nd	0.08	0.07	0.06	**Nitrogen compounds**				
1-Phenylbutene	nd	0.01	0.01	4.52					
Benzene	1.09	1.69	1.72	1.58	Unknown	0.32	0.85	1.1	0.92
**Ketones**					Benzeneacetonitrile	0.14	0.34	0.15	0.13
2-Propanone	1.48	3.13	1.6	1.07					
2,3-Butanedione	0.12	0.16	0.11	0.12	**Sulfur compounds**				
Hydroxyacetone	0.62	0.8	1.22	5.96	Dimethyl sulfide	1.70	1.35	1.81	3.24
1-Hydroxy-2-butanone	nd	0.03	0.31	0.13	Dimethyl sulfoxide	nd	0.02	0.33	0.27
Lactone	0.35	0.34	0.29	0.37	Dimethyl sulfone	nd	0.47	0.65	0.61
4-Oxoisophorone	0.26	2.24	2.28	1.83					
Unknown	nd	0.14	0.61	0.36	**Esters**				
Geranyl acetone	0.15	0.21	0.16	0.06	Ethyl acetate	nd	0.19	0.12	0.16
Pantolactone	0.09	0.1	0.12	0.14					

(a *) = Volatile compounds in honey identified at T0; (b **) = means of storage periods at different storage temperatures (ST15, ST25, ST35); Unknown = compound recognised by using NIST library for its chemical class.

**Table 3 foods-12-03105-t003:** LSMeans of VOCs classes and HMF, and related standard error (SE) at each storage temperature. Means with different superscripts significantly differ (*p* < 0.05).

Chemical Class	ST15	ST25	ST35	SE
Aldehydes (%)	2.18 ^a^	2.02 ^a^	3.46 ^b^	0.16
Alcohols (%)	44.71 ^a^	45.80 ^a^	26.86 ^b^	0.49
Hydrocarbones (%)	5.56	5.64	5.34	0.38
Ketones (%)	6.06 ^a^	4.92 ^b^	9.01 ^c^	0.27
Free Fatty Acids (%)	30.44 ^a^	26.63 ^b^	24.22 ^c^	0.59
Terpenes (%)	3.75 ^a^	4.36 ^b^	4.00 ^a^	0.18
Furans (%)	2.22 ^a^	4.89 ^b^	20.31 ^c^	0.16
Nitrogen Compounds (%)	2.34 ^a^	1.64 ^b^	1.24 ^c^	0.14
Sulfur Compounds (%)	1.85 ^a^	2.53 ^b^	4.14 ^c^	0.14
HMF (mg/kg)	5.77 ^a^	22.42 ^b^	240.24 ^c^	0.67

**Table 4 foods-12-03105-t004:** Diastase activity (DN) of the pooled honey samples at four times for each storage temperature (ST).

Time	ST15	ST25	ST35
T0	12.1	12.1	12.1
T1	11.5	11.8	8.4
T2	10.1	9.9	6.6
T4	13.2	11.3	5.1
T8	11.8	6	3.7

**Table 5 foods-12-03105-t005:** Canonical loadings, relative percentage of variance, and canonical correlations. Can1, first canonical variate; Can2, second canonical variate.

Sensor	T1		T2		T3		T4		T5		T6		T7		T8	
	Can1	Can2	Can1	Can2	Can1	Can2	Can1	Can2	Can1	Can2	Can1	Can2	Can1	Can2	Can1	Can2
2-W5S	0.63	0.12	0.16	−0.19	−0.20	0.24	−0.42	−0.15	0.13	0.02	−0.52	0.62	−0.02	0.56	0.95	0.22
3-W3C	0.57	0.16	0.20	−0.19	−0.23	0.15	−0.43	−0.04	0.23	0.00	−0.25	0.66	0.03	0.49	0.86	0.07
9-W2W	−0.75	−0.02	0.32	0.44	−0.08	−0.19	0.19	0.06	0.46	−0.06	0.93	−0.20	0.86	−0.14	−0.94	−0.21
7-W1W	−0.20	−0.08	−0.62	0.41	0.33	0.11	0.45	−0.12	−0.44	0.44	−0.11	0.57	−0.02	−0.60	−0.63	0.21
5-W5C	−0.61	−0.30	−0.39	0.46	0.60	0.05	0.74	−0.10	−0.35	0.12	0.44	−0.38	0.02	−0.52	−0.92	−0.12
10-W3S	−0.74	0.29	0.45	0.58	−0.77	−0.37	−0.83	0.36	0.82	−0.13	0.93	−0.10	0.70	−0.06	−0.88	−0.25
8-W2S	−0.60	−0.37	−0.49	0.46	0.67	0.05	0.79	−0.08	−0.54	0.03	0.00	−0.47	0.07	−0.52	−0.96	−0.19
4-W6S	−0.62	−0.10	−0.12	0.58	0.35	0.02	0.58	−0.02	0.10	0.25	0.87	0.02	0.64	−0.08	−0.91	−0.22
1-W1C	0.61	0.19	0.21	−0.17	−0.34	0.10	−0.56	0.02	0.18	0.12	−0.44	0.66	0.30	0.61	0.72	−0.10
6-W1S	−0.28	−0.26	−0.20	0.45	0.88	0.24	0.94	−0.16	−0.40	0.41	0.11	0.46	−0.03	−0.51	−0.69	0.05
Variance accounted for %	0.85	0.15	0.97	0.03	0.98	0.02	0.98	0.02	0.96	0.04	0.95	0.05	0.72	0.28	0.96	0.04
Canonical correlation	0.95	0.79	0.99	0.81	0.99	0.81	1.00	0.84	1.00	0.94	1.00	0.97	0.98	0.95	0.98	0.72

## Data Availability

The data used to support the findings of this study can be made available by the corresponding author upon request.

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
