# Peer review of "Monitoring Volatile Organic Compounds and Aroma Profile of *Robinia pseudoacacia* L. Honey at Different Storage Temperatures during Shelf Life"

_foods, 2023, doi:10.3390/foods12163105_

Round 1
Reviewer 1 Report
Sara Panseri et al. reported “MONITORING VOLATILE ORGANIC COMPOUNDS AND AROMA PROFILE OF ROBINIA PSEUDOACACIA HONEY AT DIFFERENT STORAGE TEMPERATURES DURING SHELF-LIFE”. Here are my suggestions.
1. The title format should be in lower case.
2. In the introduction section few paragraphs are very short and need to merge with each other.
3. What do you mean by high temperature (100, 200 or 500 ˚C)? Why we need to heat the honey at high temperature. I think it’s enough to heat up to 35-40 ˚C. explain it in detail in the introduction section.
4. VOCs are highly volatile compounds (can evaporate very rapidly at very lower temperature too). In the table 2, why Acetic acid concentration from 29.79 (T0) to 16.2 (T35). It’s not so prominent change because at 35 its value should be lower. Give the reason in the revised manuscript. Check all this compound values (Furan, Acetonitrile, etc.).
5. Conclusion need improvement.
6. Improve the English through English expert.
7. Check the grammatical and spelling mistakes throughout the manuscript.
Moderate editing of English language required
Author Response
Dear Reviewer 1,
Thank you for your careful revision and valuable comments. Please find thereafter our point-by-point answers. Associated changes have been tracked in red in the revised manuscript.
Sara Panseri et al. reported “MONITORING VOLATILE ORGANIC COMPOUNDS AND AROMA PROFILE OF ROBINIA PSEUDOACACIA HONEY AT DIFFERENT STORAGE TEMPERATURES DURING SHELF-LIFE”. Here are my suggestions.
1.The title format should be in lower case.
Answer: The title was modified as suggested.
2.In the introduction section few paragraphs are very short and need to merge with each other.
Answer: Some paragraphs were merged as suggested to make the introduction more fluent. Please see e.g. lines 52-56 and 60-73.
3.What do you mean by high temperature (100, 200 or 500 ˚C)? Why we need to heat the honey at high temperature. I think it’s enough to heat up to 35-40 ˚C. explain it in detail in the introduction section.
Answer: The mentioned temperatures were not used to simulate storage conditions of honey; the tested storage temperatures were representative and useful to monitor the shelf life also considering the possible abuse conditions in terms of temperature. Therefore, 35 °C was tested as maximum storage temperature to mimick suboptimal storage conditions or mild heating.
- VOCs are highly volatile compounds (can evaporate very rapidly at very lower temperature too). In the table 2, why Acetic acid concentration from 29.79 (T0) to 16.2 (T35). It’s not so prominent change because at 35 its value should be lower. Give the reason in the revised manuscript. Check all this compound values (Furan, Acetonitrile, etc.).
Answer: The volatile compounds are listed in Table 2 as average percentage during the entire storage period for each temperature, in order to present a general overview of volatile compounds changes. The concentration trend of some organic acids, such as acetic acid, may be also influenced by mild fermentation phenomena that can start at higher storage temperatures and produce new molecules of these acids; this might explain what remarked by the Reviewer 1, about the drop in acetic acid concentration not as fast as might have been expected during storage. The following brief comment at this regard has been added to the manuscript in Discussion, lines 431-435: “It should be considered that Acetic acid may be also be produced in honey as a result of mild fermentation phenomena that can start at higher storage temperatures; this may have contributed to the concentration trend observed in the present study, which declined not as fast as it might have been expected during storage.” Regarding Acetonitrile, the results of the analyses were carefully re-evaluated, and then we concluded that it was a contaminant since present in the laboratory environment. Thus, we decided to eliminated Acetonitrile from Table 2.
- Conclusion need improvement.
Answer: The Conclusions were implemented, describing in particular the advantages and limitations of the use of e-nose. Please see lines 469-479.
- Improve the English through English expert.
Answer: The manuscript was carefully checked by native English-speaking expert.
- Check the grammatical and spelling mistakes throughout the manuscript.
Answer: The manuscript was carefully checked for grammatical and spelling mistakes.

Reviewer 2 Report
This manuscript was well written with an interesting method of monitoring volatile organic compounds and aroma profile or Robinia pseudoacacia honey at different storage temperature during shelf-life and detecting aromatic alterations in honey by using electronic nose. The experiments were carried out properly, only one point should be asked of the authors. What is the storage day in each temperature (ST15, ST25 and ST35) of the data in Table 2? Moreover, it will be more interesting if the authors discuss the limitation of the use of e-nose in the manuscript.
Comments in details:
Page 2-3 line 92-100: Please check the sentence that does not correspond to the content of the specified line.
Table 1: Please check the subscript of all chemical formula.
Table 2: Please check the chemical name such as Terpen, n-compound and cinnammalcohol. In addition, please check the accuracy of the substances in the Miscellaneous group. Because some compounds can be classified as other groups, for example, thymol can be classified as terpenes.
Table 3: What is SE?
Please check reference format especially references 37 and 40.
Author Response
Dear Reviewer 2,
Thank you for your careful revision and valuable comments. Please find thereafter our point-by-point answers. Associated changes have been tracked in red in the revised manuscript.
This manuscript was well written with an interesting method of monitoring volatile organic compounds and aroma profile or Robinia pseudoacacia honey at different storage temperature during shelf-life and detecting aromatic alterations in honey by using electronic nose. The experiments were carried out properly, only one point should be asked of the authors.
What is the storage day in each temperature (ST15, ST25 and ST35) of the data in Table 2? Moreover, it will be more interesting if the authors discuss the limitation of the use of e-nose in the manuscript.
Answer: The volatile compounds presented in Table 2 were detected at Day 0 (T0); this Table also reports the average concentration of the same VOCs throughout the whole storage period for each temperature batches (ST15, ST25 and ST35, with the aim to provide an overall scenario focused on storage temperatures. At this regard, the legend of Table 2 and the relevant text in Section 3.1 was made clearer (please see lines 220-222). More specific and relevant trends during storage time were also shown in the figures. Regarding e-nose technique, it is less accurate than HS SPME and GC/MS but less expensive than the traditional analysis. It is useful, faster and does not require qualified manpower. Relevant comments have been inserted in the text as suggested, please see Conclusions lines 469-479.
Comments in details:
Page 2-3 line 92-100: Please check the sentence that does not correspond to the content of the specified line.
Answer: The sentence was eliminated since there was a mistake in its presence as suggested.
Table 1: Please check the subscript of all chemical formula.
Answer: Table 1 was carefully checked for all chemical formulas as recommended.
Table 2: Please check the chemical name such as Terpen, n-compound and cinnammalcohol. In addition, please check the accuracy of the substances in the Miscellaneous group. Because some compounds can be classified as other groups, for example, thymol can be classified as terpenes.
Answer: Table 2 was carefully checked for compounds names as well as chemical groups especially those reported in the Miscellaneous section.
Table 3: What is SE?
Answer: It is the acronym of Standard Error; for greater clarity, Standard Error was reported in full text and as an acronym (SE) in the legend of Table 3.
Please check reference format especially references 37 and 40.
Answer: All references were carefully checked.
Reviewer 3 Report
The present manuscript titled "MONITORING VOLATILE ORGANIC COMPOUNDS AND AROMA PROFILE OF ROBINIA PSEUDOACACIA HONEY AT DIFFERENT STORAGE TEMPERATURES DURING 4 SHELF-LIFE" presents a comparative study for monitoring of VOC in honey samples at different storage temperatures for different time intervals.
The manuscript falls well within the scope of "Foods". However, following issues needs to be addressed before acceptance of the manuscript:
[1] Introduction Section; Line 92-101: There is some irrelevant text which might have been copy-pasted in this manuscript. Author needs to delete this text from the manuscript.
[2] In SPME Section please mention the temperature for pre-equibration temeprature.
[3] Author did not study various stationary phases of SPME fibers, which might have some significant effect over total number of analytes extracted.
[4] Literature survey for importance of SPME in VOC analysis is poor. Author is suggested to cite following studies which are relevant to VOC analysis:
Journal of Chromatography B, 925, 63-69.
Food chemistry, 158, 497-503.
Author Response
Dear Reviewer 3,
Thank you for your careful revision and valuable comments. Please find thereafter our point-by-point answers. Associated changes have been tracked in red in the revised manuscript.
The present manuscript titled "MONITORING VOLATILE ORGANIC COMPOUNDS AND AROMA PROFILE OF ROBINIA PSEUDOACACIA HONEY AT DIFFERENT STORAGE TEMPERATURES DURING 4 SHELF-LIFE" presents a comparative study for monitoring of VOC in honey samples at different storage temperatures for different time intervals.
The manuscript falls well within the scope of "Foods". However, following issues needs to be addressed before acceptance of the manuscript:
[1] Introduction Section; Line 92-101: There is some irrelevant text which might have been copy-pasted in this manuscript. Author needs to delete this text from the manuscript.
Answer: The sentence was eliminated since it was a copy-paste error as correctly assumed by the Reviewer.
[2] In SPME Section please mention the temperature for pre-equibration temeprature.
Answer: The equilibration temperature was indicated in the SPME Section 2.2 as suggested (please see lines 133-134).
[3] Author did not study various stationary phases of SPME fibers, which might have some significant effect over total number of analytes extracted.
Answer: The use of CAR/PDMS/DVB fiber was preferred considering our previous studies on honey VOCs as well as the results reported in other researches. The fiber type as well as extraction temperature represent important variables since they can influence the obtained VOCs profile as well. CAR/PDMS/DVB fiber represents a useful tool to obtain a comprehensive wide-range VOCs profile for foods as honey and other bee products. A corresponding comment has been added to the Discussion, please see lines 358-362.
[4] Literature survey for importance of SPME in VOC analysis is poor. Author is suggested to cite following studies which are relevant to VOC analysis:
Journal of Chromatography B, 925, 63-69.
Food chemistry, 158, 497-503.
Answer: As recommended, the above citations were included in the Reference list as n. 21 and n. 20 respectively.